

# Microbial community analysis in the gills of abalones suggested possible dominance of epsilonproteobacterium in *Haliotis gigantea*

Yukino Mizutani[1], Tetsushi Mori[2], Taeko Miyazaki[1], Satoshi Fukuzaki[1] and Reiji Tanaka[1]

[1] Graduate School of Bioresources, Mie University, Tsu, Mie, Japan
[2] Department of Biotechnology and Life Science, Tokyo University of Agriculture and Technology, Koganei, Tokyo, Japan

## ABSTRACT

Gills are important organs for aquatic invertebrates because they harbor chemosynthetic bacteria, which fix inorganic carbon and/or nitrogen and provide their hosts with organic compounds. Nevertheless, in contrast to the intensive researches related to the gut microbiota, much is still needed to further understand the microbiota within the gills of invertebrates. Using abalones as a model, we investigated the community structure of microbes associated with the gills of these invertebrates using next-generation sequencing. Molecular identification of representative bacterial sequences was performed using cloning, nested PCR and fluorescence in situ hybridization (FISH) analysis with specific primers or probes. We examined three abalone species, namely *Haliotis gigantea*, *H. discus* and *H. diversicolor* using seawater and stones as controls. Microbiome analysis suggested that the gills of all three abalones had the unclassified Spirochaetaceae (one OTU, 15.7 ± 0.04%) and *Mycoplasma* sp. (one OTU, 9.1 ± 0.03%) as the core microbes. In most libraries from the gills of *H. gigantea*, however, a previously unknown epsilonproteobacterium species (one OTU) was considered as the dominant bacterium, which accounted for 62.2% of the relative abundance. The epsilonproteobacterium was only detected in the gills of *H. diversicolor* at 0.2% and not in *H. discus* suggesting that it may be unique to *H. gigantea*. Phylogenetic analysis performed using a near full-length 16S rRNA gene placed the uncultured epsilonproteobacterium species at the root of the family Helicobacteraceae. Interestingly, the uncultured epsilonproteobacterium was commonly detected from gill tissue rather than from the gut and foot tissues using a nested PCR assay with uncultured epsilonproteobacterium-specific primers. FISH analysis with the uncultured epsilonproteobacterium-specific probe revealed that probe-reactive cells in *H. gigantea* had a coccus-like morphology and formed microcolonies on gill tissue. This is the first report to show that epsilonproteobacterium has the potential to be a dominant species in the gills of the coastal gastropod, *H. gigantea*.

Corresponding author
Reiji Tanaka,
tanakar@bio.mie-u.ac.jp

## INTRODUCTION

Abalone is one of the worldwide valuable fisheries resources that is mainly consumed in East Asia such as China, Japan and South Korea. Farming abalone is particularly active in China and South Korea, and has recently increased in South Africa (*Cook, 2016*). Research on the relationship between abalone and bacteria is important for stable abalone culture. The digestive tract of abalone is more developed than bivalves, which are filter feeders, because abalones feed brown algae that contains polysaccharides such as alginates. Therefore, the studies of bacteria related to abalones are commonly focused on microbes associated to the gut (*Tanaka et al., 2004*; *Zhao et al., 2012*; *Nam et al., 2018*). However, it is still unclear on the role of microbes in other important organs within abalone.

It has been reported that some marine invertebrates have chemoautotrophic bacteria in/on their gills (*Dubilier, Bergin & Lott, 2008*). Symbiotic relationships between marine invertebrates and chemoautotrophic bacteria are characterized by the exchange of chemical molecules, such as carbon dioxide, oxygen, hydrogen, hydrogen sulfide, nitrogen or methane, from the hosts, through the seawater to symbionts. The symbionts then take up these molecules and provide their hosts with organic compounds. Therefore, it can be said that gills are also play an important for symbiotic bacteria as taking nutrients through seawater in addition to uptake oxygen for the hosts.

Since marine invertebrates with vestigial digestive tracts nutritionally depend on their symbionts (*Le Pennec, Beninger & Herry, 1995*), most of the studies on the microbial communities in invertebrate gills have focused on deep-sea floor vent and cold seep environments where there are many kinds of gutless organisms (*Dubilier, Bergin & Lott, 2008*). Alternatively, it has also been reported that bivalves in the coastal area have symbiotic bacteria in/on their gills, although the number of reports is more scarce than one related to deep-sea floor. The members of bivalves in three families, Solemyidae, Thyasiridae and Lucinidae, are known as shallow-water invertebrates having autotrophic sulfur-oxidizing bacteria in/on their gills (*Duperron et al., 2013*). These hosts have autotrophic sulfur-oxidizing Gammaproteobacteria in common, but the symbiotic location vary between the host species; the symbionts are intercellular in Solemyidae and Lucinidae (*Cavanaugh, 1983*; *Frenkiel & Mouëza, 1995*), while most members of Thyasiridae have extracellular symbionts on their gill tissue (*Dufour, 2005*). Symbiotic bacteria play a nutritionally important role in bivalves on coastal area as on deep-sea floor. *Solemya velum*, affiliated to the Solemyidae family, derive more than 97% of their carbon from symbionts, although they are capable of suspension feeding (*Conway, Capuzzo & Fry, 1989*; *Krueger, Gallager & Cavanaugh, 1992*). Additionally, symbionts associated with two species in Lucinidae, *Loripes lucinalis* and *Codakia orbicularis*, have been reported to fix atmospheric nitrogen and synthesize amino acids for their hosts in addition to carbon fixation and sulfur-oxidization (*Petersen et al., 2016*; *König et al., 2016*). Meanwhile, genomic information of symbiotic bacteria of *S. velum* suggests that they have both abilities to function as endo-symbiotic and free-living lifestyle (*Dmytrenko et al., 2014*). According to *Cúcio et al. (2018)*, draft genomes closely related to the gill symbiont of *S. velum* were retrieved from the rhizobiome of the seagrass where the clam inhabit,

although *Krueger, Gustafson & Cavanaugh (1996)* suggested that *S. velum* vertically transmit their gill symbionts to offspring. Furthermore, *C. orbiculata* can acquire symbionts from the environment even if the clams are adult (*Gros et al., 2012*). These symbiotic bacteria may have flexible lifestyles.

To date, only Gammaproteobacteria has been reported for symbionts on/in gills of shallow-water invertebrates. On the other hand, Epsilonproteobacteria are known as symbionts of invertebrates on the deep-sea floor. The deep-sea organisms such as *Petrasma* sp. (*Rodrigues et al., 2010*), *Rimicaris exoculata* (*Suzuki et al., 2005*) or *Alviniconcha hessleri* (*Urakawa et al., 2005*; *Suzuki et al., 2006*) have Epsilonproteobacteria in/on their gills that are known as ecologically significant group of bacteria in deep-sea hydrothermal vents and cold seeps (*Nakagawa et al., 2005*; *Takai et al., 2005*). Gamma-and Epsilon-proteobacteria in/on the gills have similar roles as carbon fixation and/or sulfide oxidation. Gamma-and Epsilon-proteobacterial symbionts mediate the Calvin–Benson cycle (*Stewart & Cavanaugh, 2006*; *Kleiner, Petersen & Dubilier, 2012*) and the reductive tricarboxylic acid cycle (*Takai et al., 2005*), respectively, to fix $CO_2$.

Bivalves and marine gastropods have open circulatory system containing hemolymph, which spread over their body including gills part. The hemolymph as the blood in invertebrates is responsible for the transport of nutrients and the elimination of invaders, particularly bacteria. It has been reported that bacteria were detected in the hemolymph of even healthy oysters (*Schmitt et al., 2012*; *Lokmer & Wegner, 2015*), while the hemolymph of abalone have strict immune system (*Hooper et al., 2007*).

In this study, we analyzed the microbial communities associated with the gills of economically important abalones, such as *Haliotis gigantea*, *H. discus* and *H. diversicolor*, in shallow-water environments. Additionally, this study investigated the phylogenetic characteristics and placement of the unique bacteria, the uncultured epsilonproteobacterium, in the gills of *H. gigantea*.

## MATERIALS AND METHODS

### Sample preparation and DNA extraction

Five cultured giant abalones, *Haliotis gigantea*, were collected from the Minami-ise Farming Fishery Center (Minami-ise, Mie, Japan) in October 2016. Wild abalone specimens (*H. gigantea*, *H. discus* and *H. diversicolor*) were obtained from a fish-market in Mie, Japan, from April 2017 to May 2019. The purchased abalones were collected offshore along the coast of Mie prefecture by scuba diver, and were transported in a water tank to the laboratory within 6 h of sampling.

Gill tissues from the cultured *H. gigantea* ($n = 5$) specimens collected from the farm and wild *H. discus* ($n = 3$) and *H. diversicolor* ($n = 5$) specimens collected from the fish-market were pooled into three individual tubes, one for each species (sample code: Hgig1, Hdis1 and Hdiv1). Gill tissues from other *H. gigantea* and *H. discus* specimens were prepared individually (sample code: Hgig2-6, Hdis2-4). Gut and foot tissues were also collected from all *H. gigantea* specimens, excluding Hgig1 from which no foot tissues were obtained. For all *H. discus* and *H. diversicolor* specimens, only gill tissues were used. The collected gill, gut or foot tissues were then homogenized in sterile artificial seawater

using a bead beater homogenizer (4,200 rpm, 30 s; Tietech Co., Nagoya, Japan) followed by a previously described method (*Tanaka et al., 2004*). Host tissues were removed from the homogenate by a quick centrifugation step (1 s, 8,000 g), and the supernatant was transferred to new tubes and centrifuged for 20 min at 15,000 g to recover bacterial cells.

Seawater and stone samples were collected from the shore in Minami-ise (34.333958 N 136.692204 E) within 30 m from the Minami-ise Farming Fishery Center. Fifty milliliters of seawaters were filtered by passing through 0.22 μm filter paper and resuspended in sterile phosphate-buffered saline (PBS: 130 mM NaCl, 10 mM Na$_2$HPO$_4$/NaH$_2$PO$_4$; pH 7.4) (sample code: SW). Stones were shaken in sterile PBS (sample code: ST) and bacterial cells were recovered from the SW and ST PBS samples by centrifuging for 20 min at 15,000 g. Bacterial genomic DNA from each bacterial pellet was extracted using a Promega DNA purification system (Promega Corp., Madison, WI, USA) according to the manufacturer's instructions.

## PCR amplification, illumina MiSeq sequencing and sequence processing

All gill, SW and ST samples were used for the analysis of microbial communities. The V1–V2 region of 16S rRNA genes was amplified using Ex Taq (TaKaRa Biotechnology Corp., Kyoto, Japan). The first PCR step was performed using primers 27F-mod (5′-ACACTCTTTCCCTACACGACGCTCTTCCGATCTAGRGTTTGATYMTGGCTCAG-3′) and 338R (5′-GTGACTGGAGTTCAGACGTGTGCTCTTCCGATCTTGCTGCCTCCCG TAGGAGT-3′). The amplification conditions were as follows: initial denaturation of 2 min at 94 °C followed by 24 cycles of denaturation for 30 s at 94 °C, primer annealing for 30 s at 55 °C, and primer extension at 72 °C for 30 s. The amplified PCR products were purified using a Wizard SV Gel and PCR Clean-Up System (Promega Corp., Madison, WI, USA) and used for the second PCR step, which was performed using primers with a tag sequence. After the second PCR, the products were once again purified using a Wizard SV Gel and PCR Clean-Up System before sequencing on a MiSeq platform (Illumina Inc., San Diago, CA, USA).

The raw paired-end FASTQ reads were demultiplexed using the Fastq barcode splitter (http://hannonlab.cshl.edu/fastx_toolkit/index.html) and imported into the Quantitative Insights Into Microbial Ecology 2 program (QIIME2, ver. 2019.7, https://qiime2.org/). Demultiplexed reads were quality filtered, denoised, chimera checked and dereplicated using a DADA2 denoise-paired plugin (*Callahan et al., 2016*). To be equal to sampling-depth, sequences were rarefied at 23,000 reads using qiime feature-table rarefy (*Weiss et al., 2017*). For preparing Upset diagram, sequences from *H. gigantea* and *H. discus* were also combined into one for each host species using feature-table merge plugin on Qiime2 in addition to prepare individual samples, and then the sequences were resampled at 23,000 reads using qiime feature-table rarefy plugin on Qiime2 (*Weiss et al., 2017*). Next, the align-to-tree-mafft-fasttree pipeline from the q2-phylogeny plugin was used to perform multiple sequence alignment, remove phylogenetically uninformative or ambiguously aligned sequences, and to generate unrooted and rooted phylogenetic trees (*Lane, 1991*; *Price, Dehal & Arkin, 2010*; *Katoh & Standley, 2013*). Diversity metrics

were calculated using the core-metrics-phylogenetic pipeline from the diversity plugin on QIIME2. For alpha diversity, Shannon index (*Shannon, 1948*), observed OTU number (*DeSantis et al., 2006*), Chao 1 index (*Chao, 1984*) and Good's coverage (*Good, 1953*) were calculated using the diversity alpha command on QIIME2. For PCoA plots, unweighted UniFrac data and Shannon diversity values were calculated by the core-metrics-phylogenetic pipeline on Qiime2, and they were combined and visualized using the R packages "qiime2R", "tidyverse", "phyloseq" (*McMurdie & Holmes, 2013*) and "ggplot2" (*Wickham, 2009*) in R (v3.5.0). Taxonomic assignments were performed using the qiime feature-classifier classify-sklearn on Greengenes v_13.8 (*McDonald et al., 2012*). Taxa bar plots were constructed using the plugin qiime taxa bar plot. For analysis of core or unique microbiome (*Reich et al., 2018*; *Neu, Allen & Roy, 2019*), Upset diagram was generated using R package UpSetR (version 1.3.3) with grouped next-generation sequencing (NGS) sequence data by each host species or environment (*Lex et al., 2014*). All of the data were deposited at the Sequence Read Archive (SRA) under the accession number PRJDB8953.

## Obtaining the 16S rRNA gene sequences of the uncultured epsilonproteobacterium by cloning

PCR amplification of the bacterial 16S rRNA gene was performed using primer 27F (5′-AGAGTTTGATCCTGGCTCAG-3′, *Lane, 1991*) and primer 1492R (5′-GGTTACC TTGTTACGACTT-3′, *Lane, 1991*). The bacterial 16S rRNA gene clone library was constructed using PCR amplicons obtained from Hgig1. PCR reaction mixtures contained 1 × PCR reaction buffer, 200 μM dNTP, 5 pmol of each primer, 2.5 units Ex Taq polymerase (TaKaRa Biotechnology Corp., Kyoto, Japan), and 10–100 ng of community DNA in a total volume of 50 μl. PCR reactions were performed on a thermal cycler (iCycler; Bio-Rad Laboratories, Hercules, CA, USA) using the following amplification conditions: initial denaturation of 4 min at 95 °C followed by 25 cycles of denaturation for 30 s at 95 °C, primer annealing for 30 s at 55 °C, and primer extension at 72 °C for 1.5 min. This was followed by a final extension reaction at 72 °C for 7 min. PCR product was ligated into the TOPO TA cloning vector (Invitrogen Corp., Carlsbad, CA, USA) according to the manufacturers' instructions. Ligation products were transformed into *Escherichia coli* One Shot TOP10 cells (Invitrogen Corp., Carlsbad, CA, USA) and clones were amplified by PCR using vector-specific primers. Plasmid DNA with insertions was sequenced using primer 27F. Partial sequencing was performed using the BigDye terminator cycle sequencing method, and an ABI 3730 sequencer (Applied Biosystems, Foster City, CA, USA). The resulting chromatograms of DNA sequences were examined using Chromas 2.33. Homology searches were performed using sequences and close relatives were identified using a BLAST search of the GenBank database on the National Center for Biotechnology Information website (http://www.ncbi.nlm.nih.gov/). The sequences identified as Epsilonproteobacteria by BLAST search were selected from random clones generated during cloning. The phylogenetic characterization of the selected sequence was estimated using the phylogenetic tree with representative sequence of Epsilonproteobacteria by the Maximum Likelihood method in MEGA 7.0 software (*Kumar, Stecher & Tamura, 2016*).

## Nested PCR analysis and sequencing of the uncultured epsilonproteobacterium

The nested PCR assay targeted 356 bp of the uncultured epsilonproteobacterial 16S rRNA gene. At the first PCR, bacterial universal primers 27F and 1492R were used, and then the uncultured epsilonproteobacterium-specific primers, Eps222F (5′-CGCTAAGAG ATTGGACTATAT-3′) and Eps578R (5′-GACTTAATAGGACACCTACATACC-3′) (designed in this study) were used at the second PCR. DNA samples extracted from gill, gut or foot tissue of each *H. gigantea* specimen were used as DNA templates. The first-round master mixture contained the following: primers (27F and 1492R, *Lane, 1991*) at 0.2 μM (each), 25 μl volume of EmeraldAmp® PCR Master Mix (TaKaRa Biotechnology Corp., Kyoto, Japan), approximately 50 ng of DNA, and up to 50 μl with distilled $H_2O$. Distilled $H_2O$ was used as the template for negative controls. Cycle parameters were 98 °C for 10 s; 10 cycles of 98 °C for 10 s, 55 °C for 30 s and 72 °C for 90 s; and 72 °C for 5 min. PCR reactions were performed using an iCycler (Bio-Rad Laboratories, Hercules, CA, USA). A 2.5-μl aliquot of the amplified PCR product was transferred to a new master mixture containing primers (Eps222F and Eps578R) at 0.2 μM (each) in a 25 μl volume of EmeraldAmp® PCR Master Mix (TaKaRa Biotechnology Corp., Kyoto, Japan) made up to a volume of 50 μl with distilled $H_2O$. Cycle parameters were 98 °C for 10 s; 25 cycles of 98 °C for 10 s, 57 °C for 30 s and 72 °C for 60 s; and 72 °C for 5 min. A 5-μl aliquot of the amplified PCR product was analyzed by agarose (1.5% (wt/vol) prepared in TAE buffer) gel electrophoresis and Midori Green Direct (Nippon Genetic Europe GmbH, Düren, Germany) staining, and the gel was photographed under UV light. The presence of a band at around 350 bp was interpreted as a positive result. Sequencing and phylogenetic analysis were performed as previously described in the subsection "Obtaining the 16S rRNA gene sequences of the uncultured epsilonproteobacterium using cloning method" above.

## Fluorescence in situ hybridization localization of the uncultured epsilonproteobacterium

Fluorescence in situ hybridization (FISH) was performed on the Hgig4 sample using a previously described method (*Tanaka et al., 2016*). Gill tissue was obtained from Hgig4 and placed in 75% ethanol at −30 °C before being fixed in 4% (v/v) paraformaldehyde/PBS at 4 °C overnight. Following fixation, specimens were rinsed and dehydrated in a 50%, 70%, 80%, 85%, 90%, 95% and 100% ethanol series, followed by 100% xylene. The fixed gill specimens were then embedded in paraffin and sliced into 10-μm transverse sections using a microtome (RV240, Yamato, Japan), before being placed on APS-coated microscope slides (Matsunami, Japan), and stored in slide boxes at room temperature until deparaffinization. Wax was removed, and tissue was rehydrated in a decreasing ethanol series. The sections were dewaxed and hybridized at 47 °C for 3 h using a solution containing 20% formamide, 0.9 M NaCl, 20 mM Tris–HCl (pH 7.4), 0.1% SDS and 0.5 μM FITC-labeled Eub338 probe (5′-GCTGCCTCCCGTAGGAGT-3′, *Amann, Krumholz & Stahl, 1990*) or 0.5 μM TAMRA-labeled Eps222 probe (5′-CGCTAAGAGATTGGACTATAT-3′), which was designed to specifically target the 16S rRNA gene of the uncultured

**Table 1 The relative abundance of the core or unique microbes in each host species and environmental control.**

|  | Hgig | Hdis | Hdiv | SW | ST |
|---|---|---|---|---|---|
| Core microbiome (%) | 12.6 | 36.3 | 12.2 | 44.5 | 20.7 |
| (No. of sequences) | (2,890) | (8,360) | (2,805) | (10,237) | (4,770) |
| Core microbiome in abalones (%) | 31.3 | 32.6 | 44.0 |  |  |
| (No. of sequences) | (7,191) | (7,508) | (10,113) |  |  |
| Core microbiome in environment (%) |  |  |  | 2.7 | 25.3 |
| (No. of sequences) |  |  |  | (631) | (5,809) |
| Unique microbiome (%) | 3.5 | 2.4 | 3.9 | 7.6 | 26.2 |
| (No. of sequences) | (807) | (551) | (898) | (1,759) | (6,033) |
| Others (%) | 52.7 | 28.6 | 39.9 | 45.1 | 27.8 |

**Note:**
The number of sequences is shown in parentheses.

epsilonproteobacterium (designed in this study). After hybridization, the sections were washed twice with washing buffer containing 20 mM Tris-HCl (pH 7.4), 180 mM NaCl and 0.01% SDS for 30 min at 48 °C, and the sections were then rinsed with ddH$_2$O and air-dried. An epifluorescence light microscope (Eclipse 400; Nikon Instech., Tokyo, Japan), was used to observe the stained cells.

# RESULTS

## Microbial community analysis by NGS

The microbial communities on the gills of various marine invertebrates were characterized by sequencing the V1–V2 region of the 16S rRNA gene. A total of 473,349 quality-filtered sequence reads were obtained from 13 samples (PRJDB8953). Alpha diversity index values for each sample are shown in supplementary Table 1. For all of the samples in this study, observed OTUs and Chao1 were approximately the same value, and Good's coverage value was higher than 99.99%, indicating that sequencing depth sufficient for capturing all bacterial species and for downstream analysis. We identified 322 OTUs from 15 phyla, which we taxonomically assigned with the Greengenes database v. 13_8. At the phylum level, the microbial taxonomic composition of most samples, excluding Hgig5, had a high relative abundance of Proteobacteria (67.8% on average). Hgig5 showed almost complete dominance of Spirochaetes (68.4%).

In Proteobacteria, sequence reads were related to Alphaproteobacteria, Gammaproteobacteria or Epsilonproteobacteria (Fig. 1). Alphaproteobacteria was dominant in Hgig1, Hgig5, Hdis2, Hdis4, SW and ST, (12.0–39.5%, Fig. 1). The most abundant Alphaproteobacteria in abalone were aligned with unclassified Rickettsiales (11.2% on average, two OTUs), while an unclassified Rhodobacteraceae (9.1%, one OTU) and *Nautella* sp. (12.5%, one OTU) were the most abundant Alphaproteobacteria-related sequences in the ST and SW samples, respectively. The relative abundance of Gammaproteobacteria-related sequences was more than 10% in all samples, except Hgig2. The genus *Vibrio* (Gammaproteobacteria) was observed in all samples. In addition, the relative abundance of *Vibrio* spp. (six OTUs) was

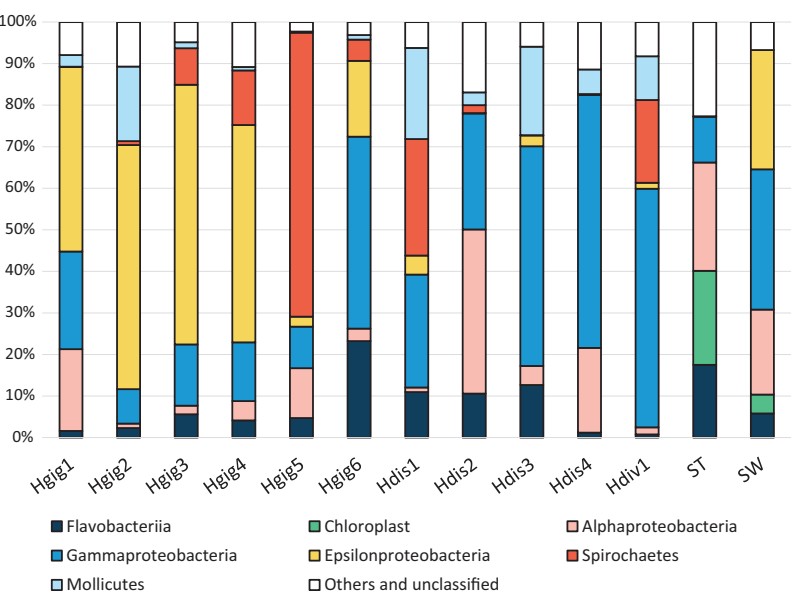

**Figure 1 Composition of the relative abundance of 16S rRNA gene amplicon reads (class-level taxa).**
16S rRNA gene amplicon reads were obtained from the gills of three abalone species, *H. gigantea* (Hgig), *H. discus* (Hdis) and *H. diversicolor* (Hdiv) and Stone (ST) and Seawater (SW). Classes with relative abundance >10% in any sample are presented. Taxa with lower abundance and unassigned taxa are indicated as "other".

more than 10% in SW and all *H. discus* specimens, especially in Hdis4 (58.4%, three OTUs). In Hdiv1, an unclassified Endozoicimonaceae (one OTU) accounted for 25.1% of the relative abundance and were more frequently recovered than *Vibrio* spp. In *H. gigantea* specimens, *Alcanivorax dieselolei* (one OTU) showed 11.8% relative abundance in Hgig1, while an unclassified Francisellaceae (one OTU) and an unclassified Gammaproteobacteria (one OTU) were distributed at 9.4% and 5.9% in Hgig6, respectively. Gammaproteobacteria-related sequences in other Hgig samples however were classified into OTUs with relative abundance less than 5%. Sulfur-oxidized Gammaproteobacteria, which was reported as symbiotic bacteria, was not detected in any samples. Epsilonproteobacteria-related sequences were dominantly recovered from SW and all *H. gigantea* specimens, except Hgig5 (18.3–62.2%, Fig. 1). All Epsilonproteobacteria-related sequences in SW were affiliated with *Arcobacter* sp. (28.7%, one OTU). *Arcobacter* sp. was also detected in the others samples, although the relative abundance was less than 5%. Other Epsilonproteobacteria-related sequences were most closely related to unclassified Epsilonproteobacteria (two OTUs). One of these was detected in all *H. gigantea* specimens and dominated in Hgig1 to Hgig4, accounting for 50.5 ± 0.05% of the relative abundance. The unclassified Epsilonproteobacteria was not found in all Hdis, SW and ST, although the related sequences represented 0.2% of the relative abundance in Hdiv1. Another unclassified Epsilonproteobacteria was detected in Hgig4, Hgig5, Hgig6, Hdis1 and Hdiv1, accounting for up to 7.1% of the relative abundance in Hgig4. Spirochaetes-related sequences were aligned with an unclassified Spirochaetaceae (one OTU). These sequences were observed in all abalone specimens except in Hdis4. The unclassified Spirochaetaceae was abundant in Hgig5, Hdis1 and Hdiv1, accounting for

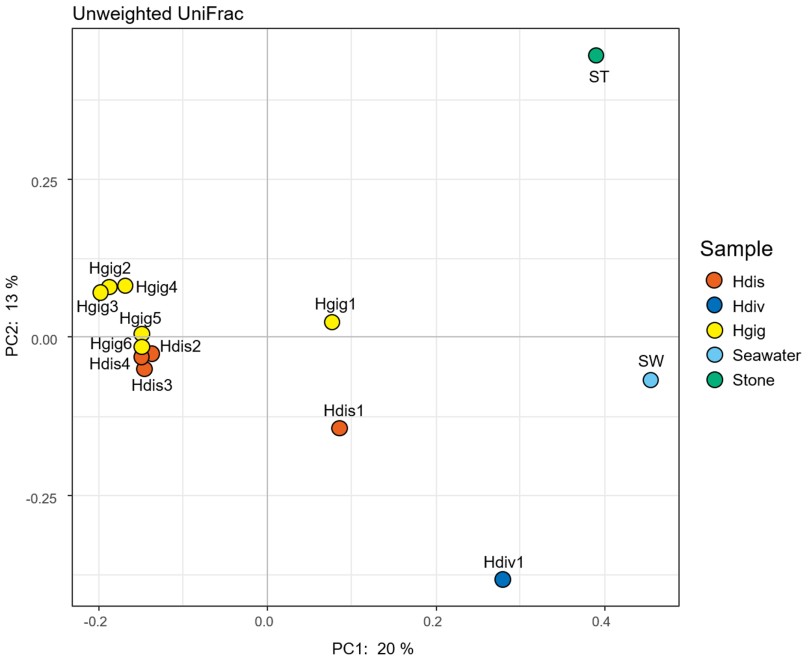

**Figure 2 Principal coordinate analysis (PCoA) of unweighted UniFrac distances for *H. discus* (Hdis), *H. diversicolor* (Hdiv), *H. gigantea* (Hgig), Seawater and Stone.**

68.4%, 28.1%, 19.9%, respectively. No Spirochaetes-related sequences were detected in SW and ST. All members of the phylum Tenericutes were affiliated with one OTU in the genus *Mycoplasma*. *Mycoplasma* sp. was present in every abalone, and most prevalent in Hdis1 (21.9%), followed by Hdis3 (21.3%). In contrast, the sequences related to *Mycoplasma* were never observed in SW and ST. Chloroplast-related sequences were only detected in ST and SW, accounting for 22.6% (six OTUs) and 4.5% (six OTUs), respectively. Most Chloroplast-related sequenceare associated with an unclassified Stramenopiles (one OTU) and accounted for up to 12.6% of the relative abundance in ST.

The similarity in the microbiota was supported by principal coordinates analysis (PCoA, Fig. 2). The microbiota in each invertebrate differed from those in the environment, such as from the seawater and stones. The microbial community structure of *H. gigantea* was similar to one of *H. discus*, while the microbe of *H. diversicolor* formed a cluster distinct from other abalone species.

The idea of core and unique microbe was used to clarify the characteristics among samples (*Reich et al., 2018*; *Neu, Allen & Roy, 2019*). The largest numbers of unique OTUs was obtained from Hgig with 64 OTUs (Fig. 3). In descending order, the unique OTUs were 52, 49 and 28 OTUs in ST, Hdis and SW, respectively. The number of unique OTUs in Hdiv was the smallest where eight OTUs were obtained. The relative abundance of the unique OTUs showed low value of 3.5%, 2.4%, 3.9% and 7.6% in Hgig, Hdis, Hdiv and SW, respectively (Table 1). On the other hand, the unique OTUs in ST showed high proportion (26.2%), although the highest relative abundance showed only 4.5%. 10 OTUs were obtained from all samples as core microbe, and the relative abundance of

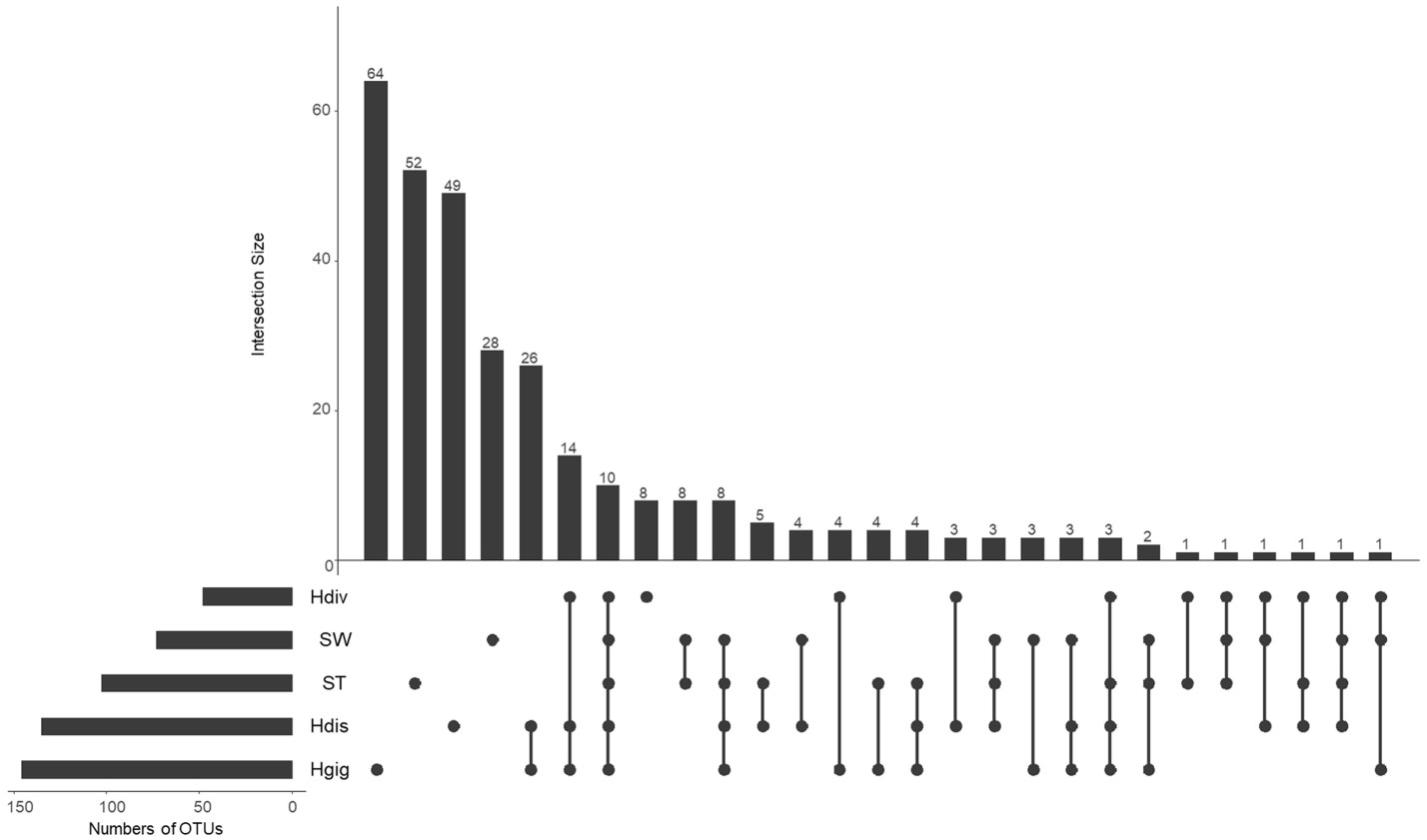

**Figure 3 UpSet plot showing shared and unique OTUs in each host species and environmental samples.** Calculations are based on OTU table rarefied to 23,000 sequences/sample. The bars show the overlap between the indicated sample below.

the core microbe showed 12.6%, 36.3%, 12.2%, 44.5% and 20.7% in Hgig, Hdis, Hdiv, SW and ST, respectively (Fig. 3; Table 1). *Arcobacter* sp. and a member of genus *Vibrio* (one OTU) were shared between all specimen as the core microbe. A total of 14 OTUs were detected as the core microbe in abalones, which was commonly obtained from Hgig, Hdis and Hdiv, and their proportions were 31.3%, 32.6% and 44.0%, in Hgig, Hdis and Hdiv, respectively (Table 1). The core microbe in abalones included the unclassified Spirochaetaceae and *Mycoplasma* sp. The core microbe in environments, SW and ST, was eight OTUs, including the unclassified Stramenopiles, and their proportions were 2.7% and 25.3% in SW and ST, respectively (Table 1).

## Phylogenetic analysis of 16S rRNA gene of the uncultured epsilonproteobacterium

Cloning and sequencing of the long 16S rRNA gene sequence of the uncultured epsilonproteobacterium found in *H. gigantea* were performed to clarify the phylogenetic position of this bacterium. Four out of 16 cloned sequences completely matched to the sequence of the uncultured epsilonproteobacterium obtained from the 16S rRNA gene amplicon sequencing undertaken in this study. The longest sequence length was 1,436 bp (LC511979).

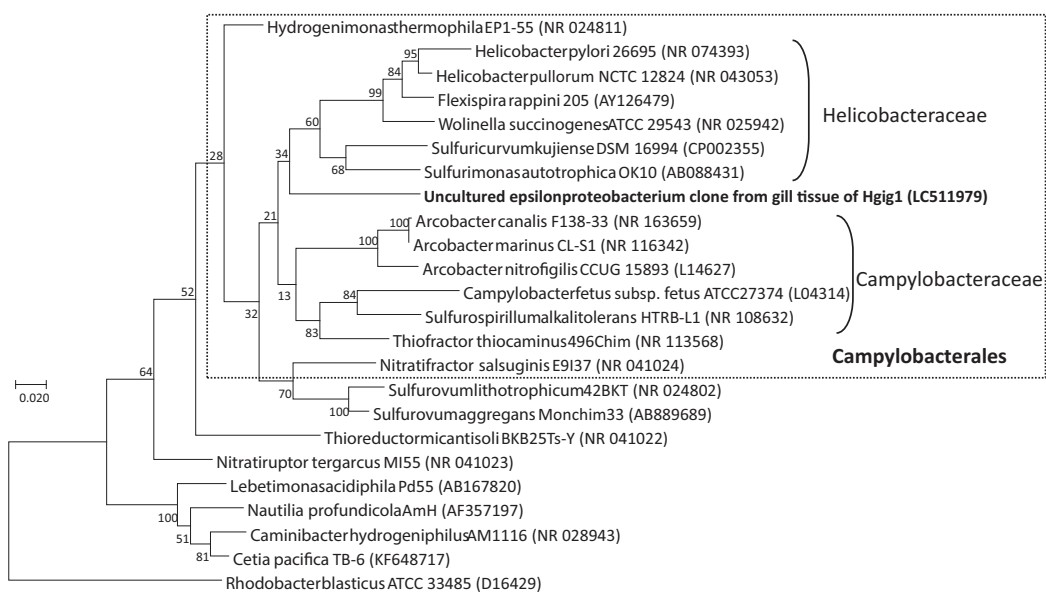

**Figure 4** **Phylogenetic tree showing the relationship between the uncultured epsilonproteobacterium and representative taxa of the Epsilonproteobacteria, based on a 1257 nucleotide sequence of the 16S rRNA gene.** The tree was inferred based on maximum likelihood. *Rhodobacter capsulatus* ATCC11166 in the Alphaproteobacteria was used as an outgroup. GenBank accession numbers are given in parentheses. Numbers at nodes are bootstrap values based on 1,000 replicates. Bar: 0.02 nucleotide substitutions per site. Taxonomic classes of the order Campylobacterales are given on the right side of round brackets.

BLAST analysis of the uncultured epsilonproteobacterial sequence revealed that the sequence could be assigned to various described Epsilonproteobacteria species, such as *Arcobacter canalis* strain F138-33 (87.43% identity), *Arcobacter marinus* strain CL-S1 (87.19% identity), *Sulfurovum lithotrophicum* strain 42BKT (87.01% identity) and *Helicobacter pullorum* strain ATCC51801 (86.78% identity), but these identity scores were low. Among the environmental clones in public databases, the uncultured epsilonproteobacterium specimen was most closely related to an uncultured clone sequence found in the gill tissue of a wood-feeding gastropod in genus *Pectinodonta* (96.12–96.78% identity, *Zbinden et al., 2010*), followed by an uncultured clone sequence from gill tissue of thyasirid bivalves from a cold seep in the Eastern Mediterranean (94.86% identity, *Brissac et al., 2011*). In the phylogenetic analysis with Maximum Likelihood method, the uncultured epsilonproteobacterium grouped within the order Campylobacterales (Fig. 4). However, the uncultured epsilonproteobacterium was not able to be classified below the order, since the phylogenetic position of the sequence was at the root of the family Helicobacteraceae.

## Detection of the uncultured epsilonproteobacterium from each part of *H. gigantea*

Results of the 16S rRNA gene amplicon sequencing analysis revealed that the uncultured epsilonproteobacterium was dominant only in the gill tissues of *H. gigantea*. Therefore, in order to determine whether the uncultured epsilonproteobacterium was restricted to

**Table 2 Detection of the uncultured epsilonproteobactrium in *H. gigantea* samples.**

| Sample code | Tissue | | |
|---|---|---|---|
| | Gill | Gut | Foot |
| Hgig1 | w | – | NT |
| Hgig2 | + | – | – |
| Hgig3 | + | w | – |
| Hgig4 | w | – | w |
| Hgig5 | – | – | – |
| Hgig6 | + | – | – |

Note:
+, positive; w, weak (i.e., PCR products obtained but the bands were very feint); –, negative; NT, not tested.

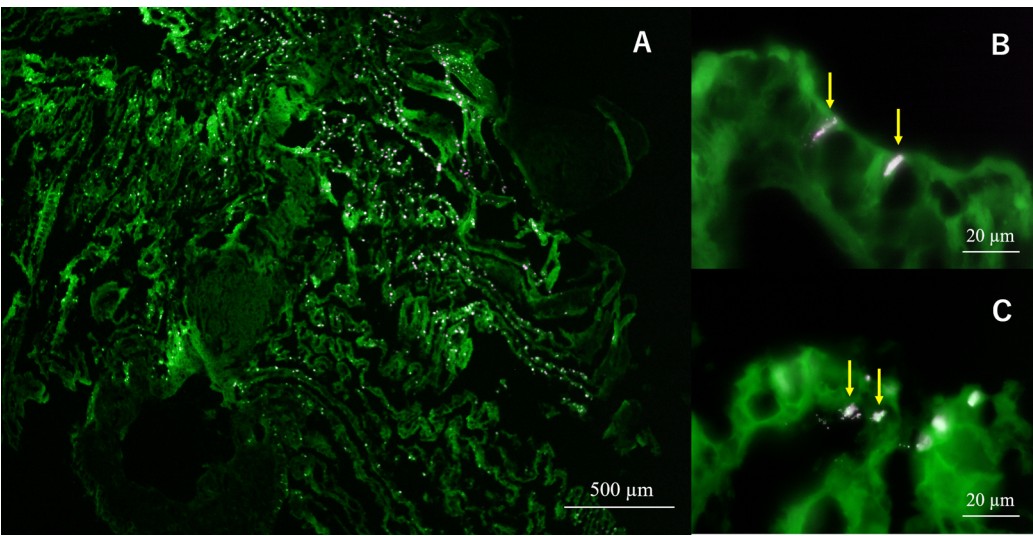

**Figure 5 Micrographs of semi-thin gill sections of *H. gigantea*-4.** Micrographs showing the results of FISH analysis of gill sample by simultaneous application of probe TAMRA-Eps222 and FITC-Eub338. (A) ×40 micrographs; (B and C) ×1,000 micrographs. White to magenta cells showed Eps222-positive cells. The yellow arrow indicates micro-colony (100–200 cells).

the gill tissues, a PCR assay of gill, gut and foot tissues from *H. gigantea* was performed using uncultured epsilonproteobacterium-specific primers. PCR products were obtained from gill tissue from Hgig2, Hgig3 and Hgig6 (Table 2). Amplified products were also obtained from the gill tissue from Hgig1, Hgig4, gut tissues from Hgig3 and foot tissues from Hgig4, but band visibility was low. All amplification product sequences matched that of the uncultured epsilonproteobacterial sequence obtained by cloning.

## Localization of the uncultured epsilonproteobacterium on gill tissue of *H. gigantea*

In semi-thin section FISH analysis, Eps222-positive bacteria were observed on the gill surface of Hgig4 (Fig. 5) where they formed microcolonies on gill tissue (Figs. 5B and 5C).

Eps222-positive bacteria showed a coccus-like morphology and measured ~1.0 μm in diameter (Fig. 5C).

## DISCUSSION

In this study, we analyzed the microbiota in the gills of three species of abalones. It was previously considered that the bacterial community in the gills would be similar to that in seawater because the gill filaments of aquatic organisms are exposed to seawater, particularly during respiration. However, PCoA analysis suggested that the microbial composition and characteristics of shellfishes differed from those of seawater and/or stone. In addition, when we look at the bacterial community structure in detail, the dominant bacterial species were different for each host species, even among congeneric hosts.

The member of the phylum Tenericutes, *Mycoplasma* sp., was detected in all abalone samples thus, classified as the core microbe in abalone. It was suggested that abalones, the genus *Haliotis*, harbor *Mycoplasma* sp. before their speciation. Additionally, the sequence of *Mycoplasma* sp. obtained in this study showed 99.7% homology identity with uncultured clones that were dominant in the gut of *H. d. hannai* (*Tanaka et al., 2004*). *Mycoplasma* sp. were also expected to inhabit not only the gill tissues, but also gut tissues. However, the associations between these potential symbiotic bacteria and their hosts is still unknown.

The sequence of the unclassified members of Spirochaetaceae in the class Spirocaetia also classified as a core microbe in abalone, were closely related to the sequence detected in the cold-water coral *Lophelia pertusa* in the northeastern Gulf of Mexico (91.2% identity, *Kellogg, Lisle & Galkiewicz, 2009*). The relationship between uncultured *Spirochaeta* and *L. pertusa* was unknown, but spirochetes have been reported to catalyze the synthesis of acetate from $H_2$ and $CO_2$ in the termite gut (*Leadbetter et al., 1999*), as well as being a symbiont of the gutless worm, *Oligochaeta loisae*, in the Australian Great Barrier Reef (*Dubilier et al., 1999*). Additionally, *Spirochaeta* symbionts are detected in the gill tissues of cold-seep clams (Lucinidae) where it is presumed that they are involved in autotrophy (*Duperron et al., 2007*). On the other hand, *Spirochaeta perfilievii* isolated from a sulfur 'Thiodendron' mat in a saline spring are able to oxidize sulfide, however, this was a heterotrophic species (*Dubinina et al., 2011*). The unclassified bacteria belonging to Spirochaetaceae in abalones may therefore be a symbiont that detoxified sulfide and/or carbon fixation.

Gammaproteobacteria was the dominant class of bacteria in *H. discus*, and most of these bacteria were *Vibrio* spp. Some members of *Vibrio* are the core microbes but were also abundant in seawater. Hence, it was considered that *Vibrio* in/on gills of abalones was infected from their surrounding environment, particularly seawater. Non-motile *Vibrio* spp., called non-motile fermenters have been reported to be symbionts of abalones (*Sawabe et al., 2003*). These authors reported that they fermented organic acids, especially acetate, from alginate of brown algae in the abalone gut and provide the resulting organic acids to their hosts. However, since it cannot be assumed that there is always alginate around the gills of abalone specimens, it is proposed that *Vibrio* spp. on the gills originated

in the gut and then adhered to the gills. Alternatively, the *Vibrio* spp. found in the gills have different ecological roles from those found in the gut.

Most symbionts in the gills of bivalve have not been isolated, and their 16S rRNA genes show very low homology with characterized species (*Dubilier, Bergin & Lott, 2008*). Therefore, these symbionts are classified as uncultured Gammaproteobacteria. We searched for these symbiont-related sequences in the uncultured Gammaproteobacteria sequences from our NGS analysis, but we unable to identify them. Since some bivalves acquire their symbionts from the environments in which they live like sediments (*Gros et al., 2012*; *Lim et al., 2019*), while abalones inhabit on reefs, we assumed that sulfur-oxidized Gammaproteobacteria was not detected in all samples.

The uncultured epsilonproteobacterium was dominant in all of the *H. gigantea* samples, except Hgig5, by NGS analysis. Interestingly, this bacterium was not dominant in the congeneric species, *H. discus* and *H. diversicolor*. The relative abundance of the uncultured epsilonproteobacterium in Hgig6 was slightly lower than that in the other *H. gigantea* samples. The high Shannon index value obtained for Hgig6 and the detection of less common bacterial species in other *H. gigantea*, such as *Polaribacter* spp. and unclassified Flavobacteriaceae species, indicates that the abundance of the uncultured epsilonproteobacterium in Hgig6 was relatively low compared to that found in other *H. gigantea* specimens.

The class Epsilonproteobacteria in terrestrial environments is widely known for its pathogenic genera; *Campylobacter*, *Helicobacter*, and to a lesser extent *Arcobacter*, (*Gilbreath et al., 2011*). Some of the members of this class also inhabit at hydrothermal vents, where they are often associated with invertebrate hosts as ecto-, endo-or epi-symbionts; for example, polychaete worm (*Alvinella pompejana*) (*Haddad et al., 1995*), shrimp (*Alvinocaris longirostris*) (*Tokuda et al., 2008*), crab (*Shinkaia crosnieri*) (*Fujiyoshi et al., 2015*) and gastropods (*Alviniconcha* spp.) (*Suzuki et al., 2005*, *2006*; *Urakawa et al., 2005*). The members of this group are considered to be symbionts that provide their hosts with organic carbon compounds by carbon fixation and detoxify hydrogen sulfide. Although the 16S rRNA gene sequence of the uncultured epsilonproteobacterium obtained from *H. gigantea* in this study formed a cluster in phylogenetic analysis with members of Campylobacterale, the similarity score to described species was as high as 87.43%. Thus, it is considered that the uncultured epsilonproteobacterium bacterium obtained in this study could be, at least, a novel genus. BLAST analysis comparing the uncultured epsilonproteobacterium with other uncultured bacterial clones revealed that they were closely related to clones detected from the gills of a wood-feeding gastropod, *Pectinodonta* sp. (Patellogastropoda, Mollusca) (96.12–96.78%, identity). Although the relationship between these clones and their host was unknown, the epsilonproteobacterial clone sequence obtained from the gills was not detected from the digestive system or foot of *Pectinodonta* sp. (*Zbinden et al., 2010*). The uncultured epsilonproteobacterium is therefore considered to specifically inhabit gill tissues.

Results of the microbial community analysis revealed that the uncultured epsilonproteobacterium was dominant in the gills of the gastropod, *H. gigantea*. However, it was surmised that the bacterium was also presents in other body parts. Like the gills, the foot of *H. gigantea* in direct contact with seawater. Compared to gills or the foot, the

gut is also a closed space in which bacteria can be easily retained. The gut environment is also microaerobic, which means that it is well suited for the growth of some members of Epsilonproteobacteria, such as *Campylobacter* and *Arcobacter* (*On et al., 2017*). Therefore, PCR using a specific primer for the uncultured epsilonproteobacterium was performed to confirm whether the bacterium also inhabited tissues other than the gills. PCR amplification products of the uncultured epsilonproteobacterium were obtained from all gill tissues from *H. gigantea*, but never or very little, from the gut and the foot tissue, even when different tissues from the same individual were used. These findings therefore suggested that the uncultured epsilonproteobacterium is generally restricted to gill tissues. In addition, semi-thin section FISH analysis revealed micro-colonies of the uncultured epsilonproteobacterium on *H. gigantea* gills. It is necessary to culture these bacteria in order to better clarify their metabolism and the biological interactions that exist between the uncultured epsilonproteobacterium and *H. gigantea*.

## CONCLUSIONS

In this study, we analyzed the microbiota that colonize the gills of abalones such as *H. gigantea*, *H. discus* and *H. diversicolor*, using 16S rRNA amplicon sequencing. The findings suggested that the gills of the abalones support specific bacterial communities compared to the environment such as seawater and stones. The unclassified Spirochaetaceae (one OTU) and *Mycoplasma* sp. (one OTU) were found in all abalone specimens as the core microbe in abalone. *H. discus* showed dominance of *Vibrio* as the core microbe in all samples including seawater and stones. An uncultured epsilonproteobacterium (one OTU) was a unique microbe in the gills of *H. gigantea*, except for detection of *H. diversicolor* (0.2%). Additionally, an uncultured epsilonproteobacterium, which dominant only in *H. gigantea*, formed micro-colonies on the gills of this species, but not in its gut or foot. This uncultured epsilonproteobacterium specifically inhabits the gills of the shallow-water gastropod, *H. gigantea*.

### Funding

This work was supported by a JSPS Research Fellowship (No. 18J14216) for Young Scientists from the Ministry of Education, Culture, Sports, Science and Technology of Japan. The funders had no role in study design, data collection and analysis, decision to publish, or preparation of the manuscript.

### Grant Disclosures

The following grant information was disclosed by the authors:
JSPS Research Fellowship: 18J14216.

### Competing Interests

The authors declare there are no competing interests.
## Author Contributions

- Yukino Mizutani conceived and designed the experiments, performed the experiments, analyzed the data, prepared figures and/or tables, authored or reviewed drafts of the paper, and approved the final draft.
- Tetsushi Mori analyzed the data, authored or reviewed drafts of the paper, and approved the final draft.
- Taeko Miyazaki performed the experiments, authored or reviewed drafts of the paper, and approved the final draft.
- Satoshi Fukuzaki analyzed the data, authored or reviewed drafts of the paper, and approved the final draft.
- Reiji Tanaka conceived and designed the experiments, analyzed the data, authored or reviewed drafts of the paper, and approved the final draft.

## Data Availability

The 16S rRNA gene sequence of the uncultured epsilonproteobacterium is available at NCBI GenBank: LC511979.

The Illumina MiSeq sequencing data are available at NCBI Sequence Read Archive (SRA): PRJDB8953.

## Supplemental Information

Supplemental information for this article can be found online at http://dx.doi.org/10.7717/peerj.9326#supplemental-information.

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
