# Peer review of "Microbial community analysis in the gills of abalones suggested possible dominance of epsilonproteobacterium in Haliotis gigantea"

_PeerJ, doi:10.7717/peerj.9326_

## Round 0.1 · original submission · Major Revisions

Although all three reviewers feel the importance of your work, BUT there need major revisions; especially reviewer 2 (Dr Nolwenn Callac) discussed very details to improve the manuscript!

Please give a second thought on Title to be precise!

·

Basic reporting

Many landmark studies on microbial community on and/or in the gill of invertebrate have been reported, so novel association is expected in marine invertebrates. Authors found unique microbial community dominated by Epsilonproteobacteria on the gill of Haliotis gigantea, and visualized those cells by whole cell in situ hybridization technique. It is very interesting study, but is needed for the publication.

Experimental design

Experiments reported in this manuscript were based on standard methodology.
However, authors only used abalone individuals form market. This might have mislead of natural microbial community of abalone.

Validity of the findings

FISH images were not sufficient to provide robust findings of presence of epsilonproteobacterial cells.

Additional comments

Many landmark studies on microbial community on and/or in the gill of invertebrate has been reported, so novel association is expected in marine invertebrates. Authors found unique microbial community dominated by Epsilonproteobacteria on the gill of Haliotis gigantea, and visualized those cells by whole cell in situ hybridization technique. It is very interesting study, but is needed for the publication.

Majors
1. Title is too long, should be simple and concise.
2. Abstract should be shortened.
3. It was wondering epsilonbacterial cells were active or not, how much sequences were matched obtained from meta16S analysis and actual single cell on the gill.
4. It was also wondering the reason why only some Haliotis gigantea individuals harbored unique Epsilonproteobacteria. Authors should confirm the presence of Epsilonproteobacteria in wild animals.
5. Authors should add anatomical description of circulation system of abalone. Authors also describe species specific features of these system, in particular in Haliotis gigantea.
6. Authors also should confirm no microbes in hemolymph of animals used in this study.
7. In fig.4, cell localization was difficult to be confirmed. Eps333 and Eub338 merged picture was necessary.
8. English must be improved.

Minors
1. L152: the subtitle should be shortened.
2. L204: a period after 80 should be changed to comma.

·

Basic reporting

Have you checked the authors data deposition statement? Yes, we can access to the data
Can you access the deposited data? Yes
Has the data been deposited correctly? Yes BUT in NCBI, the accession number corresponding to the stone sample, for the organism description it is written sediment metagenome, why? If it’s stones’ sample.
"DRX188994 Sample: Stones SAMD00192396 • DRS114354 • Organism: sediment metagenome”
Is the deposition information noted in the manuscript? Yes

Experimental design

Most of the manuscript is clear and well written in professional, unambiguous language.
The structure of the manuscript is well conforms to PeerJ standards, discipline norm, except the materials and methods section which needs to be improved in term of clarity (see comments below).

The main aim and rationale of the study need to be clearer explain as well as the focus on the specific Epsilonproteobacteria taxa.

The Figures are relevant, with high quality, well labelled & described.
Raw data supplied (see PeerJ policy) are available

Intro & background to show context. Literature well referenced & relevant.
The introduction described well the state of the art of the gills’ symbiont topic with relevant and right amount of references but in my opinion, focuses too much on deep sea hydrothermal or cold seep systems, even if the comparison between shallow and deep sea on lines 82-86 is well appropriated.
From line 67 to 71, on the Solemyidae description, it could be good to add few references about the symbiotic sulfur-oxidizing symbionts and to link that with the Thyasiridae description:
Dmytrenko, Oleg, et al. "The genome of the intracellular bacterium of the coastal bivalve, Solemya velum: a blueprint for thriving in and out of symbiosis." BMC genomics 15.1 (2014): 924 ; see also other papers
Also, the authors described the Solemyidae and the Thyasiridae and their symbionts’ roles but then, somehow it felt like the description of the species used in this study is lacking. Indeed, several information about their life’s styles, previous evidences of symbiont in the gill of specimens of the same family etc… will be beneficial for the manuscript.

Validity of the findings

The findings are well explained. Statistical analysis are done when it's possible. The conclusions are coherent with the results shown in the manuscript however some words about H. discus, H. diversicolor, T. cornutus could be a good improvement of the conclusion.

There are some discrepancy between the text and the phylogenetic tree: with the tree lines: 276-279 “BLAST analysis of the uncultured epsilonproteobacterial sequence revealed that the sequence could be assigned to various described Epsilonproteobacteria species, such as Arcobacter canalis strain F138-33 (87.43% identity), Arcobacter marinus strain CL-S1 (87.19% identity), Sulfurovum lithotrophicum strain 42BKT (87.01% identity) and Helicobacter pullorum strain ATCC51801 (86.78% identity), but these identity scores were low” BUT in the phylogenetic tree (Figure 2), clone Hgig1 is closest to Sulfurimonas autotrophica and Sulfuricurvum kujiense and bit further from Arcobacter nitrofigilis. That differs from the manuscript. Also, why not using Arcobacter canalis, Arcobacter marinus, Sulfurovum lithotrophicum, Helicobacter pullorum to do the phylogenetic tree??

Additional comments

The manuscript “Microbial community analysis in the gills of coastal shellfish and molecular identification of the potentially dominant epsilonproteobacterium on the gills of Haliotis gigantea” presents a relevant and interesting study, describing the gill bacterial composition of 4 species of gastropds (Haliotis gigantea, H. discus, H. diversicolor and Turbo cornutus) and 2 species of molluscs species (Meretrix lusoria and Cyclina sinensis) using high throughput sequencing, cloning-sequencing and microscopic methods. Besides, the authors revealed the evidence of a specific taxon of uncultivated Epsilonproteobacteria closely affiliated to Helicobacteraceae family, using specific primers designed for this purpose, in the gill of the abalone Haliotis gigantea
The manuscript flow well and it well written and the topic is really interesting.
However, there is several main points, that should be added or clarified before the paper is recommended for publication

Major comments:
The manuscript will really gain in clarity if the main aims of this study were clearly described in the introduction: why studying these specific species, why comparing molluscs and gastropods Which scientific gaps this study ambitions to explore. That will really help the readers to understand the study and its rationale.
Also, the authors should highlight why they did a specific focus on the uncultured Epsilonproteobacteria detection in different tissues of Haliotis gigantea. Is this specific taxon also present in the others abalone gills (Haliotis discus and H. diversicolor)? Did the authors checked that using the specific primers that they designed for the uncultivated Epsilonproteobacteria detected in Haliotis gigantea?

The manuscript: discussion and conclusion mostly, will gain in relevance if the authors dig deeper in the analysis of the gill microbiome, meaning not only the classes but maybe until the family or genus level when possible. For instance, the histogram shows only the classes with a relative abundance >10% and in the text, only few genera are highlighted as Vibrio, Arcobacter. Then, for example, I wonder what are their relative abundance within the Gammaproteobacteria and Epsilonproteobacteria respectively.

The manuscript will improve a lot if the authors can highlight both the core microbiome and variable microbiome, up to the genus level, in the gills of Haliotis gigantea itself and in H. diversicolor and compare and point out some similarity or dissimilarity among the core microbiote of these 2 species belonging to the same genus.
See for example: Neu, Alexander T., Eric E. Allen, and Kaustuv Roy. "Diversity and composition of intertidal gastropod microbiomes across a major marine biogeographic boundary." Environmental microbiology reports 11.3 (2019): 434-447.
Reich, Inga, et al. "16S rRNA sequencing reveals likely beneficial core microbes within faecal samples of the EU protected slug Geomalacus maculosus." Scientific reports 8.1 (2018): 1-9.
Hernandez-Agreda, Alejandra, Ruth D. Gates, and Tracy D. Ainsworth. "Defining the core microbiome in corals’ microbial soup." Trends in Microbiology 25.2 (2017): 125-140.

The material and method need some clarifications:
Does sample Hgig1 corresponds to the pool of 5 specimen collected at the farm? Not clear
“Other abalone specimens (H. gigantea, H. discus and H. diversicolor) and Turbo cornutus were obtained from a fish-market in Mie, Japan, from April 2017 to May 2019”: Any idea of where they come from? Where had they been collected?
“Gill tissues from H. discus (n=3), H. diversicolor (n=5), and T. cornutus (n=3) specimens were pooled into three tubes, one for each species (sample code: Hdis1, Hdiv1, and Tcor1)”: Can you explain why you have decided to pool the gill of each specie together?
H. discus specimens corresponding to samples Hdis2-4: I Can’t reckon where come from the sample 1 for H Discus? Line 100 H. discus n=3; line 103 sample name Hdis2-4 so no Hdis1 in the manuscript BUT in graph yes there is a sample Hdis 1 and in NCBI as well. Here some explanations are missing.
“Seawater and stone samples were collected from Minami-ise” Some indications are missing as the sampling point coordinates, when have they been collected, Which year? How have they have been collected Are these samples been collected close to the Farming Fishery Center, from the Farming Fishery Center?
Also, line 369: “sediments smelt …”why did you investigated the microbial diversity in the stones and not in the sediment and to look for potential symbionts inside?
Obtaining the 16S rRNA gene sequences of the uncultured epsilonproteobacterium by cloning:
“The phylogenetic tree of representative members of Epsilonproteobacteria inferred from 16S rRNA gene sequences was estimated by the Maximum Likelihood method using MEGA 7.0 (Kumar et al. 2016).” That’s not clear, I get that the authors wanted to obtain the almost full length of 16S rRNA gene in order to get a fair idea of the affiliation and taxonomy of this uncultured epsilonproteobacterium taxon but here I missed a link because the authors used universal primers that target all the Bacteria and not only the epsilonproteobacteria. Also, why only the epsilonproteobacteria are in the phylogenetic tree? How did the authors checked that send only Epsilonproteobactria clones to be sequenced?
Nested PCR analysis and sequencing of the uncultured epsilonproteobacterium:
“The nested PCR assay targeted 356 bp of the uncultured epsilonproteobacterial 16S rRNA gene using bacterial universal primers 27F and 1492R, as well as the uncultured epsilonproteobacterium-specific primers, Eps222F (5'-CGCTAAGAGATTGGACTATAT-3') and Eps578R (5'-GACTTAATAGGACACCTACATACC-3') (designed in this study).” Here the sentence is not clear, does that mean that the authors have used the same PCR products as for the cloning-sequencing? About the Eps222F/ Eps578R primers, more precisions are needed: are these primers been design for the NGS data, from the almost full length 16S rRNA gene? Specificity of the primers?
“Sequencing and phylogenetic analysis were performed as previously described in the subsection “Obtaining the 16S rRNA gene sequences of the uncultured epsilonproteobacterium using cloning method” above” then, why only one sequence of 1436 pb has been deposited in GenBank (LC511979.1). There is no accession number for the nested PCR data.
“Four out of sixteen cloned sequences were closely related to the sequence of the uncultured epsilonproteobacterium obtained from the 16S rRNA gene amplicon sequencing undertaken in this study. The longest sequence length was 1436 bp (LC511979).” How many clones have been sequenced? How did the authors checked they send only epsilonproteobacteria clones to be sequenced? Why only 16 sequences: it’s very few especially if the nested PCR have been done from the DNA from the gills of the 5 organisms? If the sequences were closely related, what is the percentage of similarity? Why only 1 accession number? Why only 1 sequence in the phylogenetic tree? How many clones did the authors managed to obtain per Hgig sample?

Minor comments:
Abstract:
-“Recently, it has been reported that chemosynthetic bacteria in the gills of some shallow-water bivalves have the ability to fix nitrogen carbon, and synthesize amino acids for their hosts.” This sentence is not useful as the study is not dealing with the metabolic activities of the symbionts and neither on which compounds the symbionts can supply to their hosts.
- “Microbiome analysis suggested that the gills of H. gigantea, M. lusoria and C. sinensis each have unique bacterial community structures that differ from those in the surrounding environment”; add detected between differ and those in the surrounding… In addition, some results about the microbial communities inhabiting the gills of Haliotis discus, H. diversicolor and Turbo cornutus would be appreciated.
-Few words about the core and variable microbiome within Haliotis gigantea and H. diversicolor will improve a lot the abstract.

Introduction:
The introduction described well the state of the art of the gills’ symbiont topic with relevant and right amount of references but in my opinion, focuses too much on deep sea hydrothermal or cold seep systems, even if the comparison between shallow and deep sea on lines 82-86 is well appropriated.
From line 67 to 71, on the Solemyidae description, it could be good to add few references about the symbiotic sulfur-oxidizing symbionts and to link that with the Thyasiridae description:
Dmytrenko, Oleg, et al. "The genome of the intracellular bacterium of the coastal bivalve, Solemya velum: a blueprint for thriving in and out of symbiosis." BMC genomics 15.1 (2014): 924 ; see also other papers
Also, the authors described the Solemyidae and the Thyasiridae and their symbionts’ roles but then, somehow it felt like the description of the species used in this study is lacking. Indeed, several information about their life’s styles, previous evidences of symbiont in the gill of specimens of the same family etc… will

Results:
“Greengenes database”: Which version

“These sequences were observed in all abalone specimens, but were not dominant, except in Hgig5 (68.4%).”, any idea why Hgig 5 is so different from the others

“PCoA”: Utilization of which software, R? Need to be described in the Materials & Methods

Discrepancy between the text and the phylogenetic tree: with the tree lines: 276-279 “BLAST analysis of the uncultured epsilonproteobacterial sequence revealed that the sequence could be assigned to various described Epsilonproteobacteria species, such as Arcobacter canalis strain F138-33 (87.43% identity), Arcobacter marinus strain CL-S1 (87.19% identity), Sulfurovum lithotrophicum strain 42BKT (87.01% identity) and Helicobacter pullorum strain ATCC51801 (86.78% identity), but these identity scores were low” BUT in the phylogenetic tree (Figure 2), clone Hgig1 is closest to Sulfurimonas autotrophica and Sulfuricurvum kujiense and bit further from Arcobacter nitrofigilis. That differs from the manuscript. Also, why not using Arcobacter canalis, Arcobacter marinus, Sulfurovum lithotrophicum, Helicobacter pullorum to do the phylogenetic tree??

Discussion:
“This assumption was also supported by the finding that the gills of Hgig6 gave a clearer band than the gills of Hgig4 in the PCR assay using the same concentration of DNA template.” qPCR is needed to infer the number of representatives of this specific taxon in each gill’ abalone.

References:
Newton ILG, et al., is followed by Katoh K et all and Kuwahara H, et al is followed by On S et al., : Check this order.

Wickham, 2016 in the manuscript while it’s 2009 in the ref Wickham H. 2009. ggplot2: Elegant Graphics for Data Analysis. New York: Springer. DOI: 10.1007/978-0-387-98141-3 in the reference list

Figure 2: species name in italic

·

Basic reporting

The manuscript informs bacterial community compositions in bivalve and gastropods, particularly epsilonproteobacterium in Haliotis gigantea. Several investigation approaches were performed to elucidate the bacterial communities including 16S rRNA amplicon sequencing, PCR and fluorescence In-situ hybridization (FISH). The manuscript is written in good English language even though some modification is needed. It also conforms to PeerJ standard including accession of DNA sequence and data. However, introduction and background of the study need to be reformulated clearly. The authors mention epsilonproteobacterium in the title; but the importance of this bacterial group is missing in the introduction section. The authors also mentioned bacterial community in stone but they did not discuss why and how the information from that sample influence/support the objective. Furthermore, figure quality and tables need to be improved.

Experimental design

Despite of varied methods and data, I observed an essential issue on sampling design. It is still not clear whether H. gigantea specimens were the same origin/nature. It is mentioned that the H. gigantea specimens were collected from farming fisheries center and fish markets. However, it is important to clarify if they are wild specimens or domesticated origin. In addition, it is also important to mention how long the specimens stayed in the fish markets which may affect bacterial community compositions in H. gigantea. Furthermore, there is an unequal samples if Hgig1 (cultured giant abalone, N=5) is a pool samples, while other Hgig samples (Hgig2-6) were prepared individually. In my opinion, this will affect 16S rRNA amplicon sequencing analyses. Please also see detail information which I provide separately (together with annotated PDF file in last page) .

Validity of the findings

The results have been described in a descriptive way. The authors conclude the study based on the result that that they have and it is in accordance to their objective.

Additional comments

The study deliver a promising result about bacterial community compositions in shallow water bivalves and gastropods. Please find my detail comment in the last page of PDF file!

---

## Round 0.2 · accepted · Accept

Thank you for your in-depth analysis on the microbial diversity in the gills of abalone and find out Epsilonproteobacteria as dominant group by using modern techniques! hope the results will be helpful for the researcher of abalone aquaculture and health management!

·

Basic reporting

this is second review, all points raised in the first review were fixed.

Experimental design

this is second review, all points raised in the first review were fixed.

Validity of the findings

this is second review, all points raised in the first review were fixed.

Additional comments

this is second review, all points raised in the first review were fixed.

·

Basic reporting

The manuscript entitled “Microbial community analysis in the gills of abalones 1 suggested possible dominance of epsilonproteobacterium in Haliotis gigantea” presents an interesting study based on various approaches: NGS, cloning-sequencing, FISH microscopy in the aim to investigate the microbial diversity inhabiting invertebrates’ gills. The authors have highlighted that the microbial diversity inhabiting the gills of Haliotis gigantean is dominated by an unknow OUT affiliated to the Epsilonproteobacteria (Helicobacteracea) while this OUT in not or very little detected in the other abalone species studied. The FISH analysis done with specific primers proved that the uncultivated Epsilonproteobacteria cells are located on the gills. The authors have also exhibited that core microbiome co-own between the 3 abalone species is relatively restraint with only 2 commons OTUs, 1 affiliated to the Spirochaetaceae family and 1 to Mycoplasma genera.
This study is relevant for the both the environmental and aquaculture field. Overall it is a very interesting study, well written which can be published.

Experimental design

Wild and cultured abalones were used for this study and molecular approach was used to describe the microbial community in the gills, foot and gut. The specimen replicates varied between 3 and 5 allowing statistical analysis. Classical methods were used: NGS, cloning-sequencing and FISH microscopy.

Validity of the findings

The results have been well described and the findings have been well explained and discussed when possible. Statistical analyses were done when it was possible. The conclusion is coherent with the results shown in the manuscript and fit the objectives of this study.

Additional comments

This version of the manuscript is clearer especially the materials and methods. Also, the objective of the study is now well described. By removing the data about the 2 molluscs species Meretrix lusoria and Cyclina sinensis, the authors allowed the manuscript to be sharper and to bring out the main conclusion about the Espilonproteobacteria dominance in the gill of H. gigantea.
In fine, this study brought new highlights into the microbial symbiosis in the gills of the coastal abalone.

There are few minor comments to fix:
Line 80 add “role” between important and for
Line 108-109 : should be “Solemya velum, affiliated to the Solemyidae family”
Line 152 : should be Fifty milliliters of seawater were filtered instead of was filtered
Line 411-412 : “The genus Vibrio (Gammaproteobacteria) was observed in all samples” instead of were
Line 452 : “The unclassified Spirochaetaceae was abundant in” instead of abundance
Line 553: “Four out of sixteen cloned sequences were completely matched” remove were
Line 670-671 : should be “where they are often associated with invertebrate hosts”
Line 698 : “gigantea in in direct” remove one in